# Oxidative Glucose Metabolism Promotes Senescence in Vascular Endothelial Cells

**DOI:** 10.3390/cells11142213

**Published:** 2022-07-16

**Authors:** Leonie K. Stabenow, Darya Zibrova, Claudia Ender, Dario L. Helbing, Katrin Spengler, Christian Marx, Zhao-Qi Wang, Regine Heller

**Affiliations:** 1Institute of Molecular Cell Biology, Jena University Hospital, Friedrich Schiller University Jena, 07745 Jena, Germany; leonie.stabenow@med.uni-jena.de (L.K.S.); daria.zibrova@uni-jena.de (D.Z.); claudia_ender@hotmail.de (C.E.); dario.helbing@leibniz-fli.de (D.L.H.); katrin.spengler@med.uni-jena.de (K.S.); 2Leibniz Institute on Aging-Fritz Lipmann Institute, 07745 Jena, Germany; christian.marx@zepai.de (C.M.); zhao-qi.wang@leibniz-fli.de (Z.-Q.W.)

**Keywords:** aging, endothelial cell, replicative senescence, glucose metabolism, lactate, lactate dehydrogenase, pyruvate dehydrogenase kinase, dichloroacetate

## Abstract

Vascular aging is based on the development of endothelial dysfunction, which is thought to be promoted by senescent cells accumulating in aged tissues and is possibly affected by their environment via inflammatory mediators and oxidative stress. Senescence appears to be closely interlinked with changes in cell metabolism. Here, we describe an upregulation of both glycolytic and oxidative glucose metabolism in replicative senescent endothelial cells compared to young endothelial cells by employing metabolic profiling and glucose flux measurements and by analyzing the expression of key metabolic enzymes. Senescent cells exhibit higher glycolytic activity and lactate production together with an enhanced expression of lactate dehydrogenase A as well as increases in tricarboxylic acid cycle activity and mitochondrial respiration. The latter is likely due to the reduced expression of pyruvate dehydrogenase kinases (PDHKs) in senescent cells, which may lead to increased activity of the pyruvate dehydrogenase complex. Cellular and mitochondrial ATP production were elevated despite signs of mitochondrial dysfunction, such as an increased production of reactive oxygen species and extended mitochondrial mass. A shift from glycolytic to oxidative glucose metabolism induced by pharmacological inhibition of PDHKs in young endothelial cells resulted in premature senescence, suggesting that alterations in cellular glucose metabolism may act as a driving force for senescence in endothelial cells.

## 1. Introduction

The endothelium plays an important role in maintaining vascular homeostasis by controlling the balance between vasodilators and vasoconstrictors, anti- and pro-inflammatory factors and antioxidants and oxidants [1]. This homeostatic function is declining with increasing age, which may ultimately lead to age-associated cardiovascular diseases [2]. Increasing evidence suggests that endothelial senescence driven by oxidative stress and chronic inflammation promotes endothelial dysfunction in aging [3,4]. Senescent endothelial cells have been shown to accumulate in normal arterial tissue during the lifespan of humans [5,6,7] and to occur in cardiovascular-disease states, such as atherosclerosis [8] and hypertension [9].

In general, senescent cells are characterized by growth arrest due to an upregulation of p53/p21 and p16/retinoblastoma gene product pathways, by profound genomic and epigenomic alterations, by dysregulation of signaling pathways and by the acquisition of an inflammatory secretory phenotype, which may spread senescence via autocrine and paracrine effects [4,10]. Senescence is likely to be interlinked with metabolic alterations. It may be driven by the disruption of metabolic homeostasis, for instance, by reduced cytosolic NAD+/NADH ratio, high glucose or insufficient autophagy, and is, on the other hand, characterized by intracellular metabolic reprogramming, which may then promote senescence in feedback loops [11]. For instance, a shift towards glycolysis and diversion of pyruvate away from oxidative metabolism, changes in lipid metabolism, including lipid droplet formation and enhanced β-oxidation of fatty acids, as well as an increase in mitochondrial and lysosomal mass associated with organelle dysfunction, have been observed in senescent cells [11,12,13].

Shifts in glucose metabolism towards glycolytic or oxidative energy production, i.e., metabolization of glucose via pyruvate to either lactate or acetyl-CoA, are controlled by the pyruvate dehydrogenase complex (PDC) [14]. This complex is regulated through the phosphorylation status of its E1-α subunit, the pyruvate dehydrogenase (PDH), which is mediated by PDH kinases (PDHKs) and PDH phosphatases, respectively. Phosphorylation of PDH at serine residues Ser232, Ser293 and Ser300 by one of the four existing PDHK isoforms leads to a strong reduction in PDC activity [15], with the consequence that pyruvate is mainly converted to lactate by the enzyme lactate dehydrogenase (LDH). In contrast, decreased phosphorylation of PDH promotes oxidative glucose metabolism in the tricarboxylic acid (TCA) cycle and the respiratory chain. The increased activity of PDH may play a role in senescence since it has recently been shown to mediate oncogene-induced senescence [16].

Senescent endothelial cells have hardly been characterized metabolically; moreover, it is still unclear whether metabolic changes may drive senescence in these cells. Quiescent endothelial cells gain most of their energy from glycolysis, i.e., from the conversion of glucose via pyruvate to lactate, despite the fact that oxidative glucose metabolism via acetyl-CoA oxidation in the TCA cycle yields significantly more ATP per molecule glucose [17,18]. Glycolytic energy production may have the advantage of maintaining endothelial function under hypoxic conditions, keeping ROS levels low, producing the angiogenic molecule lactate and generating building blocks for macromolecule synthesis. Accordingly, glycolytic flux is upregulated in proliferating endothelial cells and angiogenesis [19] but an increased mitochondrial energy metabolism seems to be required for these processes as well [20,21]. In senescent endothelial cells, glycolysis is shown to decline [22] or not to be changed [23] and mitochondrial metabolism remains largely unexplored. Here, we provide a detailed analysis of glucose metabolism in replicative senescent versus young endothelial cells. Our data reveal that senescent endothelial cells increase their glycolytic and mitochondrial energy production, which is associated with an increased expression of lactate dehydrogenase A (LDHA) and a decreased expression of all four PDHK isoforms, respectively. Pharmacological inhibition of PDHKs by dichloroacetate (DCA) [24], resulting in the activation of PDH, inhibition of glycolysis and redirection of pyruvate towards the TCA cycle, induces a premature senescence phenotype, indicating the importance of metabolic reprogramming in endothelial senescence and vascular ageing.

## 2. Materials and Methods

### 2.1. Chemicals

Medium 199 (M199) was purchased from Lonza (Verviers, Belgium) and fetal calf serum (FCS), human serum and endothelial growth supplement (ECGS) were obtained from Sigma (Taufkirchen, Germany). Collagenase was obtained from CellSystems^®^ GmbH (Troisdorf, Germany). The protease inhibitor mixture complete, EDTA-free, was obtained from Roche Diagnostics (Mannheim, Germany). Heparin, glutamine, penicillin/streptomycin, trypsin/EDTA (0.05%/0.02%), vitamin C, DCA, 5-bromo-4-chloro-3-indolyl-β-D-galactopyranoside (X-gal), the Glucose (GO) Assay Kit, 2,2-azino-bis(3-ethylbenzothiazoline-6-sulfonic acid) diammonium salt (ABTS), lactate standard for ion chromatography, lactate oxidase, horseradish peroxidase (HRP), o-dianisidine dihydrochloride, dimethylsulfoxide (DMSO), sodium pyruvate, oligomycin, carbonyl cyanide-4-(trifluoromethoxy)phenylhydrazone (FCCP), antimycin A, 2-deoxyglucose, N,N,N′,N′-tetramethyl ethylenediamine (TEMED), staurosporin and sodium dodecyl sulfate (SDS) were purchased from Sigma (Taufkirchen, Germany). Radioactive-labeled D-glucose [1-^14^C or 6-^14^C] was obtained from Hartmann Analytic GmbH (Braunschweig, Germany). MitoSOX™ Red and Mitotracker Green FM were derived from Thermo Scientific (Waltham, MA, USA). Human serum albumin (HSA) was purchased from Bayer Vital GmbH (Leverkusen, Germany). Methanol and calcium chloride was derived from AppliChem GmbH (Darmstadt, Germany). The sources of other employed materials and test kits are mentioned in the respective method sections.

### 2.2. Antibodies

Rabbit monoclonal antibodies against β-actin, PDHK1 (C47H1), LDHA (C4B5), p21 (12D1) and the polyclonal antibody against vinculin were obtained from Cell Signaling Technology (Frankfurt, Germany). Mouse monoclonal antibodies against PDHK2 (5F8), PDHK3 (2B11) and the rabbit polyclonal antibody against PDHK4 (RB03041) were from Novus Biologicals (Centennial, CO, USA). The rabbit polyclonal antibody against p-PDH-E1α (pSer293) (AP1062) was obtained from Merck Chemicals GmbH (Darmstadt, Germany), and the mouse monoclonal antibody against PDHE1α was obtained from Santa Cruz Biotechnology (Dallas, TX, USA). A monoclonal antibody against p16 was purchased from BD Biosciences (Heidelberg, Germany), and a polyclonal antibody against p16 (ab189034) was obtained from Abcam (Cambridge, UK). Peroxidase-labeled anti-mouse and anti-rabbit IgG were from Kirkegaard and Perry Laboratories, Inc. (Gaithersburg, MD, USA).

### 2.3. Cell Culture

Experiments were performed in human umbilical vein endothelial cells (HUVEC), isolated from anonymously-acquired umbilical cords according to the “Ethical principles for Medical Research Involving Human Subjects” (Declaration of Helsinki 1964) as previously described [25]. The study was approved by the Jena University Hospital ethics committee. Donors were informed and gave written consent. Briefly, endothelial cells were detached with collagenase (0.01% in M199, 3 min at 37 °C), suspended in M199/10% FCS, washed once (centrifugation at 500× *g* for 6 min), resuspended and seeded on culture flasks coated with 0.2% gelatine. After 24 h, the medium was aspirated, full growth medium was added (M199, 17.5% FCS, 2.5% human serum, 7.5 µg/mL ECGS, 7.5 U/mL heparin, 680 µM glutamine, 100 µM vitamin C, 100 U/mL penicillin, 100 µg/mL streptomycin) and cells were cultured until they reached confluence. Then, cells were detached by trypsin/EDTA (3 min, 37 °C), split and reseeded. Senescent cells were obtained by the serial passaging of primary cells. Half of each passage was frozen in liquid nitrogen. For experiments, primary cells and cells from the 19th -24th passages were thawed to obtain 1st (P1) and 20th–25th (P20–25) passage cells (15–18 cumulative population doublings), which were grown to confluency. For individual experiments, senescent cells of one distinct passage between P20 and P25 were used and compared to P1 cells (young cells). Most experiments were performed with cells on 30-mm dishes. For ATP and glucose flux measurements, 24-well plates were used. Real-time metabolic analysis using the Seahorse XF96 extracellular flux analyzer (Agilent Technologies, Santa Clara, CA, USA) was performed on 96-well Seahorse XF96 microplates. DCA (2–20 mM, 24 h) or FCCP (5 µM, 6 days, daily readdition in fresh medium) treatments of cells were carried out in full growth medium, which was freshly added to the cells 2 days after seeding, if not otherwise indicated.

### 2.4. Cell Lysis and Western Blot

Endothelial cells were lysed in a cold Tris buffer (50 mM Tris (pH 7.4), 2 mM EDTA, 1 mM EGTA, 50 mM NaF, 10 mM Na_4_P_2_O_7_, 1 mM Na_3_VO_4_, 1 mM DTT, 1% Triton X-100, 0.1% SDS, 1 mM PMSF,10 μL/mL protease inhibitor cocktail) for 15 min on ice and scraped. After centrifugation of lysates (700× *g*, 6 min), aliquots of the supernatants were used for protein determination according to Lowry with bovine serum albumin (BSA) as standard (DCTM Protein Assay kit, Bio-Rad Laboratories, Inc., Hercules, CA, USA). The remaining supernatants were mixed with Laemmli buffer, boiled and kept at −20 °C. SDS polyacrylamide gel electrophoresis (SDS-PAGE) was performed with 20–30 µg lysate protein per lane (equal amounts per experiment) and followed by the transfer of proteins to PVDF membranes. The membranes were blocked for 1 h in a TBST buffer (20 mM Tris (pH 7.6), 137 mM NaCl, 0.1% (*w*/*v*) Tween^®^ 20) containing 5% BSA or 5% non-fat dried milk, dependent on the antibodies to be used. The blots were incubated overnight at 4 °C with primary antibodies and, subsequently, with secondary antibodies for 1 h. Antibodies against PDHK2, PDHK3 and PDHK4 and the secondary antibodies were diluted in TBST with 5% milk, and all other antibodies were diluted in TBST containing 5% BSA. For the visualization of protein signals, an enhanced chemiluminescence (ECL) reagent (GE Healthcare, Chicago, IL, USA) or a Western Lightning Plus-ECL reagent (Perkin Elmer, Waltham, MA, USA) was applied. The obtained blots were quantified with Fiji [26]. The figures represent protein bands from the same blots.

### 2.5. Senescence-Associated β-Galactosidase (SA-β-gal) Staining

Cells were washed twice with cold phosphate-buffered saline (PBS) and fixed with 2% formaldehyde/0.2% glutaraldehyde for 3 min. SA-β-gal staining was performed as previously described [27]. Briefly, after two washing steps with PBS, cells were incubated at 37 °C overnight in a dry incubator (no CO_2_) with a staining solution (40 mM citric acid/Na phosphate buffer, 5 mM K_4_[Fe(CN)_6_]·3H_2_O, 5 mM K_3_[Fe(CN)_6_], 150 mM NaCl, 2 mM MgCl_2_, 1:20 X-gal (20 mg/mL in dimethylformamide) in water, pH 6.0). Pictures were taken (EVOS™ FL Auto Cell Imaging System, Thermo Scientific, Waltham, MA, USA). SA-β-gal-positive cells were counted using Fiji [26] and normalized to the total cell number.

### 2.6. Glucose Consumption

Cellular glucose consumption was calculated from the difference in glucose levels in medium incubated with and without cells for 24 h. As medium, M199 without phenol red containing 2% FCS and 2 mM glutamine was used and either added to cells or cell-free dishes. If the effect of DCA was studied, the incubation was performed in the presence and absence of 2–20 mM DCA. After 24 h, the medium was removed from dishes and stored at −20 °C. Cells were trypsinized and counted for the normalization of glucose values. To measure glucose, the colorimetric Glucose (GO) Assay Kit (Sigma, Taufkirchen, Germany) was applied according to the manufacturer’s instructions. In this test, glucose oxidation is coupled to the formation of a colored product, oxidized o-dianisidine, whose intensity at 540 nm is proportional to the glucose concentration in the medium. The assay mix contained a 100 µL diluted sample (1:20), or glucose standard and 200 µL assay reagent, including o-dianisidine and glucose oxidase/peroxidase. The reaction was performed in 1.5 mL Eppendorf tubes for 30 min at 37 °C and under gentle shaking and stopped with 200 µL 12 N H_2_SO_4_. Samples were then transferred to 96-well plates (100 µL per well, triplicates). The analysis was performed using a SunriseTM microplate reader (Tecan GmbH, Crailsheim, Germany) and the Magellan 6 software.

### 2.7. Lactate Production

Lactate production was measured in cell supernatants after 24 h of the incubation of cells in M199 without phenol red containing 2% FCS and 2 mM glutamine. If appropriate, 2–20 mM DCA were added during this incubation. A colorimetric detection assay was applied, in which lactate oxidation led to the subsequent oxidation of the chromogenic substrate ABTS to a colored dye, whose intensity is measured at 405 nm [28]. Samples (triplicates) or lactate standards (3 µL) were added to a 197 µL assay buffer (0.1 M citric acid, 0.1% CaCl_2_, pH adjusted to 6.0 with 1 M Na_2_HPO_4_) containing lactate oxidase (0.12 U/mL), HRP (0.4 U/mL) as well as ABTS (0.02%). After incubation in 96-well plates for 30 min in the dark, absorbance was measured with a SunriseTM microplate reader (Tecan GmbH, Crailsheim, Germany; Magellan 6 software). Lactate values were normalized to the respective cell numbers per dish, which were counted after the trypsinization of cells.

### 2.8. ATP Assay

For sample preparation, cells on a 24-well plate were denaturated by adding 250 µL 96% ethanol and, after evaporation of the ethanol, 100 μL Tris-EDTA buffer (pH 7.5). After a freezing/thawing cycle in liquid nitrogen, cells were scraped and centrifuged, and the supernatants were used for ATP measurements in duplicates. In parallel, cells of a second 24-well plate treated on par with the cells of the first plate were lysed with a solubilization buffer (100 mM NaOH, 1.9 M Na_2_CO_3_, 1% SDS). The protein content of lysates was determined according to Lowry and used for the normalization of ATP values. To detect intracellular ATP levels, the ATP Kit SL (BioThema, Handen, Sweden) was performed according to the manufacturer’s instructions. The light emission of the ATP reagent SL was measured using an Autolumat LB953 (Herthold Technologies, Bad Wildbach, Germany) and calibrated with an internal ATP standard.

### 2.9. Seahorse Real-Time Cell Metabolic Analyses

Cells were seeded on Seahorse XF96 Cell Culture microplates to confluence (4000 cells/well) and were subjected to metabolic profiling in sextuplicate two days after seeding. For Mito stress tests, the growth medium was replaced with Seahorse XF Base Medium (Agilent Technologies, pH adjusted to 7.4), supplemented with 10 mM D-glucose, 2 mM L-glutamine, 1 mM sodium pyruvate and 0.25% human serum albumin (HSA). Cells were then cultured for 1 h in a CO_2_-free incubator at 37 °C. Oxygen consumption rates (OCR) were monitored at basal conditions and, after sequential injections of 2 µM oligomycin to block the mitochondrial ATP synthase, 2 µM FCCP to uncouple oxidative phosphorylation and 2 µM antimycin A was used to fully inhibit mitochondrial respiration. For glycolysis stress tests, the growth medium was replaced with Seahorse XF Base Medium supplemented with 2 mM L-glutamine, and cells were cultured for 1 h in a CO_2_-free incubator at 37 °C. Extracellular acidification rate (ECAR) was monitored at basal conditions and after sequential injections of 10 mM D-glucose, 2 µM oligomycin and 50 mM 2-deoxyglucose, an inhibitor of glycolysis. Measurements of OCR and ECAR were performed in a 3 min mix and 3 min measure cycles at 37 °C on a Seahorse XFe96 Analyzer (Agilent Technologies). Wave software (Agilent Technologies) was used to analyze the datasets. OCR and ECAR were represented as pmol/min and mpH/min, respectively, and normalized to the exact cell number of each well measured by high-content microscopy. To this end, cells were fixed for 10 min with 100% methanol at room temperature, washed once with PBS and incubated for 10 min with 1 µg/mL DAPI. After two more washing steps with PBS, cell nuclei were counted on an ImageXpress Micro confocal high-content imaging system (Molecular Devices, San Jose, CA, USA).

### 2.10. Glucose Flux Measurement

To monitor the flux of glucose through the TCA cycle and the pentose phosphate pathway (PPP), the trapping of ^14^CO_2_ released from cells incubated with [6-^14^C]-glucose (TCA cycle only) or [1-^14^C]-glucose (PPP and TCA cycle) was applied. Each flux experiment was paralleled by an identically handled plate without the addition of radioactivity, which was used for cell lysis with 200 µL solubilization buffer, protein determination according to Lowry and normalization of the flux data to the respective protein content of cells. Cells seeded on 24-well plates in duplicates were kept in M199, including 0.25% HSA for 5 h, before 1 μCi of [1-^14^C]-glucose or 1 µCi [6-^14^C]-glucose was added. Wells were covered with Whatman filter paper (2.1 × 2.1 cm) and fixed with surgical tape for a dense closure. After 2 h at 37 °C, filter papers were damped with 200 μL 3 M NaOH and cells were denaturated by the addition of 30 μL 70% perchloric acid. The plate was covered with a lid and kept overnight at 4 °C. Thereafter, filter papers were transferred to a scintillation vial and incubated for 10 min in a 2 mL scintillation cocktail. Finally, radioactivity (cpm) was measured in a liquid scintillation counter (LSC, Wallac 1410 WinSpectral, Perkin Elmer, Waltham, MA, USA) for 1 min per sample. A negative control containing only filter paper and aliquots of each radiolabeled glucose (0.2 μCi) pipetted on filter papers were included. The latter served to calculate nmoles of released CO_2_ from the respective cpm values. CO_2_ release from senescent cells was depicted as percentage of release from young cells.

### 2.11. Mitochondrial ROS Measurement

To detect mitochondria-derived reactive oxygen species (ROS), endothelial cells were stained with MitoSOX™ Red, a mitochondrial superoxide indicator. Cells were washed with PBS before 600 µL of the staining solution containing 3 µM MitoSOX™ was added. After 30 min incubation at 37 °C, cells were rinsed with PBS, detached with 300 µL trypsin/EDTA and transferred to 1.7 mL HEPES/FCS (10 mM HEPES (pH 7.4), 145 mM NaCl, 5 mM KCl, 1 mM MgSO_4_, 1.5 mM CaCl_2_, 10 mM glucose, 10% FCS). Cells were centrifuged (500× *g*, 1 min), resuspended in 300 µL PBS and subjected to flow cytometry analysis (20,000 events per sample, phycoerythrin channel, FACSCanto flow cytometer, BD Biosciences, Heidelberg, Germany). Median values were evaluated using the FlowJo™ software 7.6.5 (FlowJo, Ashland, OR, USA). Data obtained in senescent cells were depicted as percentage of data from young cells.

### 2.12. Detection of Mitochondrial Mass

Endothelial cells were washed with PBS and incubated with 600 µL of pre-warmed 50 nM MitoTracker™ staining solution for 30 min at 37 °C. Cells were then rinsed with PBS, trypsinized, transferred to 1.7 mL HEPES/FCS, centrifuged (500× *g*, 1 min) and resuspended in 300 µL PBS. Samples were analyzed by flow cytometry (20,000 events per sample, Alexa Fluor 647 channel, FACSCanto flow cytometer, BD Biosciences, Heidelberg, Germany). Median values of MitoTracker™-positive cells were analyzed using the FlowJo™ software.

### 2.13. Statistical Analysis

Each analysis was performed in 3–8 independent experiments, in which endothelial cells from different donors were used, respectively. Statistical analyses were performed using Graphpad Prism 8.4.3 (Graphpad Software, San Diego, CA, USA). A Shapiro–Wilk normality test was carried out before further statistical analyses were conducted. Depending on the result, parametric (two-tailed Student’s *t*-tests) or non-parametric tests (a Mann–Whitney test and a Wilcoxon matched-pairs signed-rank test) were performed, as stated in the figure legends. To test for equality of variances for unpaired, parametric analyses, F-tests were conducted. If an F-test showed a significant difference between two datasets, a Student’s *t*-test with Welch’s correction was performed. For measurements, in which experimental samples were normalized to the respective controls (P1 cells or unstimulated control cells), a one-sample *t*-test against 100% was performed. For dose–response experiments with DCA, the following methods were applied: For normally distributed data, a parametric repeated measures one-way ANOVA with post-hoc Holm-Šídák test of all groups against the control group was conducted. For not-normally distributed data, the non-parametric equivalent, the Friedman test with a post-hoc Dunn’s test of all groups against the control group was conducted. All data are presented as mean + SD. Statistical significance was accepted if *p* < 0.05 (depicted as one asterisk). All figures were created using Graphpad Prism 8.4.3 or Microsoft PowerPoint and assembled in Microsoft PowerPoint.

## 3. Results

### 3.1. Long-Term Culture Leads to Replicative Senescence of Endothelial Cells

HUVEC reached a state of cellular senescence after 20–25 passages (15–18 cumulative population doublings). This was verified by an increase of cells with positive staining for the senescence-associated β-galactosidase (SA-β-gal) activity (Figure 1A,B) and by the upregulation of the senescence markers p16 (cyclin-dependent kinase inhibitor 2A) and p21 (cyclin-dependent kinase inhibitor 1A) (Figure 1C,D) after long-term passage compared to young endothelial cells.

### 3.2. Replicative Senescent Endothelial Cells Exhibit Increased Glycolytic Activity

To characterize energy metabolism in replicative senescent endothelial cells, we first looked into glucose consumption and the fermentative branch of glycolysis. Replicative senescent endothelial cells consumed about 80% more glucose within 24 h than young cells (Figure 2A). In parallel, lactate production was increased by 60% indicating an increase in energy production via anaerobic glycolysis (Figure 2B). In line with this, higher intracellular ATP levels were observed in senescent endothelial cells (Figure 2C). To further substantiate these data, we measured glycolytic function in young and senescent cells using the Glycolysis stress test of the Seahorse system. Here, ECAR related to the release of lactate and protons into the extracellular medium was sequentially monitored under glucose-free conditions, after the addition of glucose, after the inhibition of the mitochondrial ATP synthase by oligomycin and, finally, after the addition of 2-deoxyglucose to inhibit glycolysis. The obtained ECAR values were used to calculate basal glycolytic activity, glycolytic capacity and glycolytic reserve. Consistent with the described increase in lactate production, replicative senescent endothelial cells showed a tendency to higher basal glycolysis (Figure 2D) and a significant increase in glycolytic capacity by approximately 30% when compared to young cells (Figure 2E) whereas glycolytic reserve remained unchanged (Figure 2F). This demonstrates the ability of senescent HUVEC to respond to increased energy demands by enhancing their glycolytic flux.

### 3.3. Replicative Senescent Endothelial Cells Show Increased Oxidative Phosphorylation

We next applied the Mito stress test of the Seahorse system to evaluate mitochondrial respiration by monitoring the oxygen consumption rate (OCR) under basal conditions and after applying specific inhibitors of the respiratory chain (rotenone, antimycin A and oligomycin to inhibit complex I, III and V, respectively) or agents to uncouple the proton gradient (FCCP). From the measured OCR values, basal and maximal respiration, respiratory spare capacity and mitochondrial ATP production were determined. Compared to young cells, replicative senescent endothelial cells showed a significant increase in basal respiration by about 20% (Figure 3A). Furthermore, mitochondrial ATP production was enhanced by approximately 30% (Figure 3B) and maximal respiration by about 40% (Figure 3C) while respiratory spare capacity was only slightly elevated (Figure 3D).

As a second approach to evaluate oxidative glucose metabolization, we measured the intracellular glucose flux using either [1-^14^C]- or [6-^14^C]-labeled glucose. While [1-^14^C]-glucose turnover monitors glucose oxidation in PPP and TCA cycle, the flux of [6-^14^C]-labeled glucose allows to assess TCA cycle activity alone. We found that the replicative senescent endothelial cells produced more ^14^CO_2_ than young cells from both precursors (Figure 3E,F). This was due to an enhanced TCA cycle activity shown as an increase in [6-^14^C]-glucose-derived ^14^CO_2_ by about 40% (Figure 3F) while glucose metabolization in the PPP as evaluated by the [1-^14^C]/[6-^14^C] ratio was not changed (Figure 3G).

Together, these data reveal an increase in oxidative glucose metabolization as a feature of replicative senescent cells. In line with this, we found an increase in mitochondrial biogenesis in senescent cells by approximately 50%, in comparison to young cells as verified by MitoTracker Green staining (Figure 3H). Increased mitochondrial metabolism was paralleled by a robust increase in mitochondrial superoxide anion production, indicating mitochondrial dysfunction in senescent cells (Figure 3I).

### 3.4. PDHK Isoforms 1–4 Are Downregulated in Replicative Senescent Endothelial Cells

Oxidative glucose metabolism is promoted by PDC, which catalyzes the irreversible decarboxylation of pyruvate to acetyl-CoA. This complex is inhibited by the action of PDHKs, which phosphorylate PDH, the E1-α subunit of PDC. To understand whether an increased glucose oxidation in replicative senescent cells is related to the altered expression of regulatory enzymes, we performed immunoblotting experiments. We found that all four PDHK isoforms were downregulated in replicative senescent cells when compared to young cells (about 50%, 70%, 30% and 40% inhibition for PDHK1, 2, 3 and 4, respectively) (Figure 4A–D). This was accompanied by a decreased phosphorylation of PDHE1-α at Ser293, indicating a higher activation of the PDC complex, which is likely responsible for the increased TCA cycle activity in replicative senescent endothelial cells (Figure 4E). The expression of PDHE1-α was not altered. In addition to decreased PDHK expression, the expression of the LDHA subunit was significantly increased (Figure 4F), which is likely to mediate the increased lactate production observed in senescent endothelial cells.

### 3.5. Inhibition of PDHKs by DCA Enhances Oxidative Glucose Metabolism and Induces Premature Senescence

To test whether a shift towards oxidative energy metabolism and the development of senescence in endothelial cells are causally related, we applied DCA, a xenobiotic pyruvate analog, which inhibits all PDHK isoforms [24,29]. We confirmed that DCA-treated endothelial cells did not show signs of apoptosis (Appendix A). Incubation of endothelial cells with 2–20 mM DCA for 24 h led to a dose-dependent decrease in PDHE1α phosphorylation at Ser293, which approves inhibition of the responsible PDHKs (Figure 5A,B). In parallel, a dose-dependent reduction in lactate secretion into the medium was observed although glucose consumption was enhanced (approximately 30% and 70% increase after 10 mM or 20 mM DCA, respectively) (Figure 5C,D). The latter may help to balance energy production since cellular ATP formation was maintained in DCA-treated cells (Figure 6A). Inhibition of glycolysis by DCA, as indicated by reduced lactate production, was confirmed by the Glycolysis Stress Test of the Seahorse system. Basal glycolysis and maximal glycolytic capacity were significantly decreased by 20 mM DCA while the glycolytic reserve was unchanged (Figure 6B–D).

Basal OCR as monitored by the Seahorse Mito stress test was unchanged in response to DCA (Figure 6E), but maximum OCR (Figure 6G) and the respiratory spare capacity (Figure 6H) were increased by about 70% and 220%, respectively. In addition, mitochondrial biogenesis was amplified by 40% (Figure 6I), pointing to an enhanced oxidative metabolism in DCA-treated cells. However, mitochondrial ATP production was not altered (Figure 6F), which may be due to insufficient energy production related to mitochondrial dysfunction. In line with this, an increase in mitochondrial ROS production up to 80% was observed (Figure 6J).

The shift away from glucose fermentation towards oxidative metabolism in DCA-treated cells was accompanied by the development of a senescent phenotype. Endothelial cells treated with 10 mM or 20 mM DCA for 24 h showed an up to 3-fold increase in SA-β-gal-positive cells (Figure 7A,B). In addition, the senescence marker p21 was dose-dependently upregulated when cells were treated with 2–20 mM DCA while p16 showed no increase under these conditions (Figure 7C,D). Together, these data suggest that increased oxidative metabolism in concert with mitochondrial dysfunction triggers premature senescence in endothelial cells. To prove this hypothesis by a different approach, we applied FCCP, a mitochondrial uncoupling agent, to increase the flux of pyruvate into mitochondria and thus oxidative glucose metabolism. As expected, treatment of endothelial cells with FCCP (5 µM, 6 days) induced premature senescence in endothelial cells as shown by a 3-fold increase in SA-β-gal-positive cells, and an upregulation of the senescence markers p21 and p16 (Figure 8A–D).

## 4. Discussion

Endothelial senescence has been implicated in the development of age-associated cardiovascular diseases but metabolism in senescent cells and the question of whether metabolic changes may drive senescence in these cells, have hardly been investigated. Here, we show that replicative senescent endothelial cells upregulate energy production from glucose, including both anaerobic glycolysis and oxidative phosphorylation. This was accompanied by an enhanced expression of LDHA and a downregulation of PDHK isoforms 1–4, the enzymes, which normally prevent oxidative metabolization of pyruvate. In addition, we provide evidence that modifying endothelial energy metabolism by inhibition of PDHKs leads to premature senescence, indicating that metabolic dysfunction drives senescence.

The upregulation of anaerobic glycolysis in our model was confirmed by different parameters, including higher glucose consumption, enhanced lactate production and increases in basal glycolytic activity and glycolytic capacity as monitored in Seahorse analyses. Our data are in line with earlier studies, showing a glycolytic state in most senescent phenotypes [12,30,31]. In contrast, previous studies in senescent endothelial cells did not show changes in the glycolytic machinery [23] or revealed even a decrease in glycolytic activity [22]. The latter was linked to a reduced expression of nuclear factor E2-related factor 2 (NRF2), a transcriptional regulator of the glycolysis-stimulating enzyme 6-phosphofructo-2-kinase/fructose-2,6-biphosphatase 3 (PFKFB3) [22]. Our data suggest that the observed increase in anaerobic glycolysis was linked to an increased expression of LDHA [32], which is known to drive generation of lactate from pyruvate and regeneration of NAD+, thereby avoiding an arrest of glycolysis due to product accumulation and low intracellular pH [33].

It is still unclear why senescent cells upregulate glycolysis. The glycolytic state has often been linked to an increased need for macromolecule building blocks required for senescence-associated events, such as senescence-associated secretory phenotype (SASP), cell enlargement or increased oxidative and endoplasmic reticulum stress, and has been thought to compensate for reduced ATP production caused by mitochondrial dysfunction [16,31,34]. In our study, increased glycolysis was not accompanied by enhanced glucose metabolization via the PPP, which branches after the first step of glycolysis and generates NADPH, a major player in redox homeostasis, and ribose 5-phosphate, a precursor for many biomolecules, including DNA and RNA. Thus, enhanced glucose consumption in senescent cells may not be connected to redox regulation or nucleotide synthesis. However, stimulation of glycolysis may contribute to the higher ATP generation observed in senescent endothelial cells although we cannot discriminate between the contribution of glycolytic metabolism and increased respiration to cellular ATP in our study.

Our data show that replicative senescent cells additionally upregulate oxidative glucose metabolism as demonstrated by two experimental approaches. We were able to detect increased TCA cycle activity by performing flux measurements using radioactive-labeled glucose, and we measured enhanced basal and maximal respiration as well as higher mitochondrial ATP production by monitoring the oxygen consumption in Seahorse analyses. These data point to an unexpected enhancement of mitochondrial function in senescent endothelial cells since, so far, senescent phenotypes are often characterized by mitochondrial dysfunction and the deterioration of oxidative phosphorylation [13]. The observed increase in respiratory function may be related to an enhanced activity of PDH, which can be concluded from its decreased phosphorylation at Ser293 and which allows an increased shuttling of pyruvate to the TCA cycle via acetyl-CoA. The decreased phosphorylation of PDH is likely attributed to the reduced expression of the responsible upstream kinases, PDHK1-4, in senescent endothelial cells. In line with this, a previous study reported that an increase in mitochondrial respiration through the activation of PDH was required for the execution of oncogene-induced senescence in primary fibroblasts and that senescence was reversed by blocking metabolic rewiring towards mitochondrial oxidative metabolism [16]. Our data suggest that mitochondrial oxidative phosphorylation may also play a role in triggering replicative senescence in endothelial cells. This may not be linked to insufficient mitochondrial ATP production, since ATP levels were enhanced in our model, may be related to redox stress caused by increased respiration. We found a substantially higher production of mitochondrial ROS in replicative senescent endothelial cells, which may trigger cellular senescence by inducing a DNA damage response, including the activation of the cell cycle inhibitors p21 and p16 [35]. Mitochondrial ROS are also known to target mitochondria themselves, thereby amplifying dysfunctions of these organelles [36]. This may result in a compensatory increase in mitochondrial biogenesis [37] while, at the same time, mitophagy is inhibited, leading to the accumulation of damaged organelles [38]. Both processes may contribute to the increased mitochondrial mass as observed in replicative senescent endothelial cells in our study.

To understand whether the activation of PDH, as seen in replicative senescent cells, is causally related to senescence, we employed a pharmacological approach. We treated early-passage endothelial cells with DCA, a well-known PDHK inhibitor and investigational drug for the treatment of metabolic diseases and cancer [29]. DCA acts via binding to the pyruvate-binding pocket of PDHK or to its allosteric site [39,40]. It is most active against PDHK2, equipotent against PDHK1 and PDHK4, and shows low activity against PDHK3 [29]. In cultured cells, including endothelial cells, DCA is usually added at mM concentrations [41,42,43,44]. DCA dose-dependently inhibited PDHK activity in endothelial cells in our study as proven by the reduction in PDH phosphorylation at Ser293 in DCA-treated cells. As expected, reactivation of PDH by PDHK inhibition led to the redirection of glucose metabolism from cytoplasmic glycolysis to mitochondrial oxidation. We found a clear inhibition of glycolysis and glycolytic capacity in Seahorse analyses, which was paralleled by reduced lactate production. Basal respiration was not changed but maximal respiration and respiratory spare capacity were significantly enhanced indicating that DCA-treated cells can upregulate oxidative phosphorylation upon demand. DCA-treated cells showed higher mitochondrial ROS production and mitochondrial mass pointing to organelle dysfunction, but mitochondrial ATP production and cellular ATP levels were maintained, possibly due to a compensatory increase in glucose consumption. These data are in line with findings from cancer cells [45], and complete a previous study in HUVEC [44]. In the latter study, the shift to oxidative metabolism by DCA was associated with the inhibition of endothelial cell proliferation and migration, but senescence parameters were not investigated [44]. Here, we show that inhibition of PDHK by DCA and the subsequent metabolic alterations promote premature senescence in endothelial cells as characterized by an increase in SA-β-gal-positive cells and by a dose-dependent upregulation of the senescence marker p21. Thus, our data reveal a shift to oxidative metabolism as a pro-senescence factor and confirm previous data on the role of PDH in regulating oncogene-induced senescence [16] although we cannot exclude an additional role of the observed inhibition of glycolysis by DCA. The latter may not only be related to increased pyruvate flux to oxidative metabolism but also to a reduced NAD ^+^/NADH ratio due to enhanced NAD^+^ consumption via the PDH-catalyzed reaction and to insufficient NAD^+^ regeneration. DCA did not alter LDHA expression as observed in replicative senescence (Appendix A). The importance of oxidative metabolism as a driver of premature senescence was confirmed in experiments, in which pyruvate flux into the respiratory chain was induced by mitochondrial uncoupling using FCCP.

Together, our data show that senescent endothelial cells upregulate both glycolysis and oxidative glucose metabolism, via an altered expression of key enzymes in these pathways, i.e., the upregulation of LDHA and the downregulation of PDHK1-4. This is associated with increased glucose consumption and higher mitochondrial and cellular ATP levels. Interestingly, stimulation of glycolysis occurs despite the fact that the mitochondrial gatekeeper PDH is activated in senescent cells; therefore, more pyruvate is metabolized into acetyl-CoA and less likely into lactate. One explanation could be that due to the increased expression of LDHA NAD^+^ is adequately regenerated to allow maintenance or even an increase in glycolysis. In addition, senescence may affect other key regulators of glycolysis and the increased glucose consumption in these cells may add to increased energy generation via two metabolic pathways. Senescence in endothelial cells is associated with an increased mitochondrial ROS production, which may potentiate this process. Increased respiration due to PDH activation is likely to promote ROS production and to trigger senescence while increased glycolysis may play a role in maintaining the senescent state, which needs to be further clarified. Our study reveals the PDHK/PDH axis as an important regulatory hub in mediating senescence, which may be used to prevent vascular senescence and ageing.

## Figures and Tables

**Figure 1 cells-11-02213-f001:**
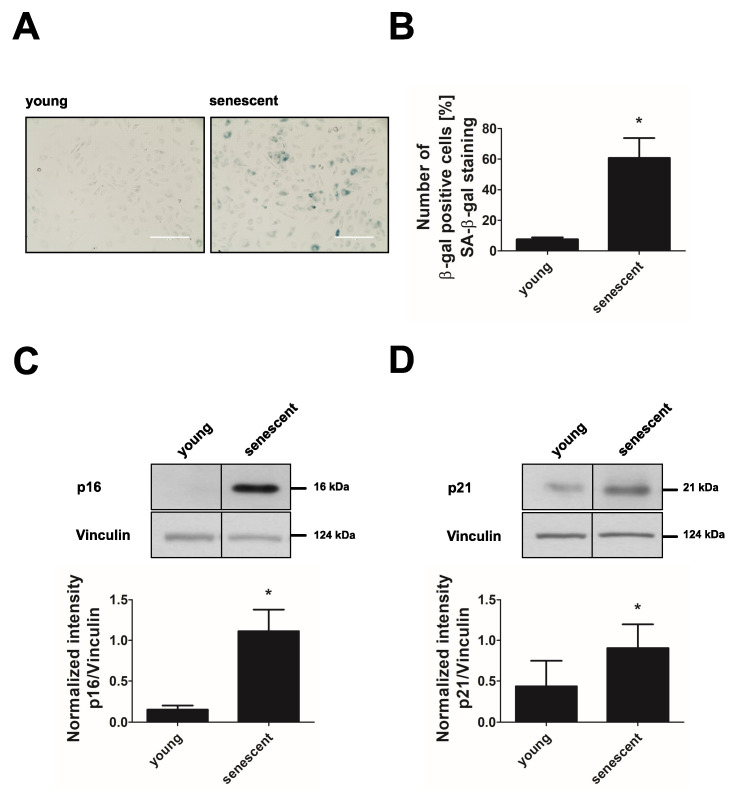
Replicative senescence in endothelial cells. HUVEC of passage 1 (young) and passage 20–25 (senescent) were subjected to analysis of senescence markers. (**A**) Senescence-associated-β-galactosidase (SA-β-gal) staining. Representative picture out of *n* = 3, scale bar = 200 µm. (**B**) Quantification of SA-β-gal-positive cells (*n* = 3, * *p* < 0.05, unpaired Student’s *t*-test with Welch’s correction). (**C**,**D**) Representative western blots of senescence markers p16 and p21 (upper panels) and densitometric quantification of p16 and p21 after normalization to vinculin (lower panels, *n* = 5, * *p* < 0.05, for p16: unpaired Student’s *t*-test with Welch’s correction; for p21: unpaired Student’s *t*-test). All data represent mean + SD.

**Figure 2 cells-11-02213-f002:**
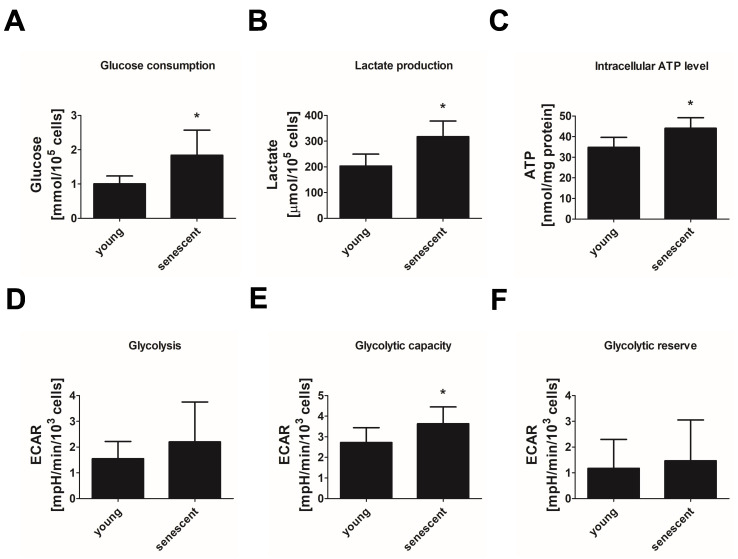
Increased glycolytic activity in replicative senescent endothelial cells. Parameters of glycolysis were analyzed in HUVEC of passage 1 (P1, young) and passage 20–25 (P20–25, senescent). (**A**) Glucose consumption as calculated from glucose levels in medium incubated for 24 h in the presence or absence of cells (*n* = 5 (P20–25), *n* = 6 (P1), * *p* < 0.05, unpaired Student’s *t*-test). (**B**) Lactate levels in medium after 24 h of incubation with cells (*n* = 5 (P20–25), *n* = 6 (P1), * *p* < 0.05, unpaired Student’s *t*-test). (**C**) Intracellular ATP level (*n* = 4, * *p* < 0.05, unpaired Student’s *t*-test). (**D**–**F**) Glycolysis stress test: glycolytic activity (**D**), capacity (**E**) and reserve (**F**) as calculated from extracellular acidification rates (ECAR) measured via Seahorse technology after sequential addition of glucose, oligomycin (inhibitor of ATP synthase) and 2-deoxyglucose (inhibitor of glycolysis) (*n* = 6 (P20–25), *n* = 8 (P1), * *p* < 0.05, unpaired Student’s *t*-test; for glycolytic activity unpaired Student’s test with Welch’s correction). All data represent mean + SD.

**Figure 3 cells-11-02213-f003:**
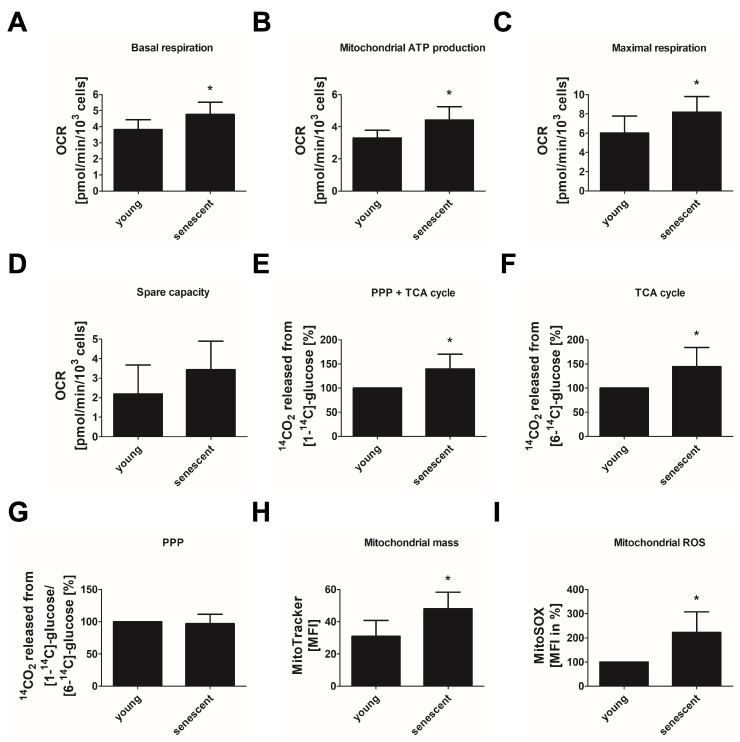
Increased oxidative phosphorylation in replicative senescent endothelial cells. Oxidative glucose metabolism and mitochondrial parameters were analyzed in HUVEC of passage 1 (P1, young) and passage 20–25 (P20–25, senescent). (**A**–**D**) Mito stress test: basal respiration (**A**), mitochondrial ATP production (**B**), maximal respiration (**C**), and spare capacity (**D**) as calculated from oxygen consumption rates (OCR) measured via Seahorse technology at baseline and after sequential addition of pharmacological agents (oligomycin, an inhibitor of ATP synthase; carbonyl cyanide-4-(trifluoromethoxy)phenylhydrazone (FCCP), a mitochondrial uncoupling agent; antimycin A, an inhibitor of the respiratory chain complex III) (*n* = 6 (P20–25), *n* = 8 (P1), * *p* < 0.05, unpaired Student’s *t*-test). (**E**) Tricarboxylic acid (TCA) cycle and pentose phosphate pathway (PPP) activity as calculated from the amount of ^14^CO_2_ released from [1-^14^C]-glucose. (**F**) TCA cycle activity as calculated from the amount of ^14^CO_2_ released from [6-^14^C]-glucose. (**G**) PPP flux depicted as ratio between ^14^CO_2_ released from [1-^14^C]-glucose and ^14^CO_2_ released from [6-^14^C]-glucose. All radioactive values were normalized to the amount of proteins of equally treated samples. Data obtained for P20–25 are shown as percentage of P1 data (*n* = 6, * *p* < 0.05, one-sample *t*-test). (**H**) Mitochondrial mass depicted as relative median fluorescence intensity (MFI) of MitoTracker staining analyzed by flow cytometry (*n* = 5, * *p* < 0.05, unpaired Student’s *t*-test). (**I**) Mitochondrial reactive oxygen species (ROS) displayed as relative MFI of MitoSOX analyzed by flow cytometry and normalized to P1 control (*n* = 5, * *p* < 0.05, one-sample *t*-test). All data represent mean + SD.

**Figure 4 cells-11-02213-f004:**
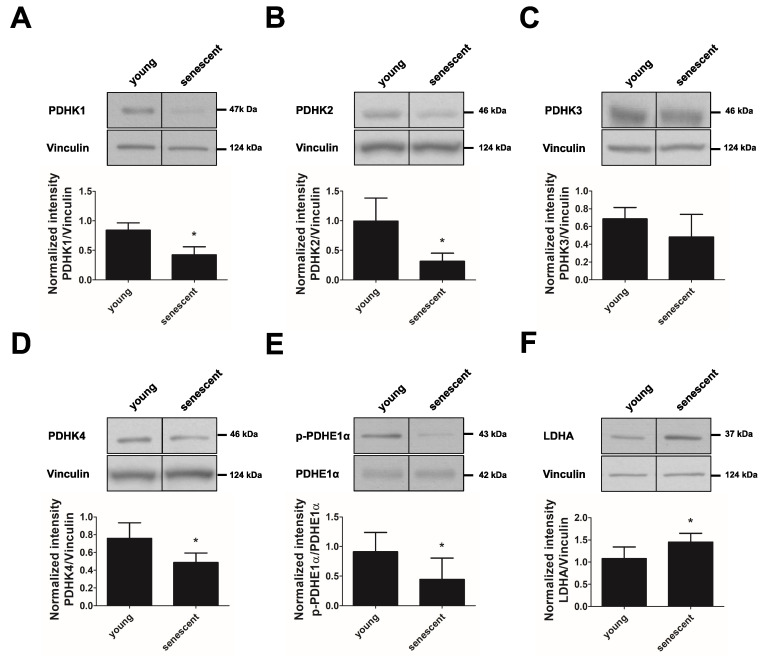
Expression and phosphorylation of metabolic enzymes in replicative senescent endothelial cells.HUVEC of passage 1 (P1, young) and passage 20–25 (P20–25, senescent) were subjected to western blot analysis of pyruvate dehydrogenase kinases isoforms 1–4 (PDHK1–4), pyruvate dehydrogenase, subunit E1*α* (PDHE1*α*), phosphorylated PDHE1*α* (at Ser293) and lactate dehydrogenase A (LDHA). (**A**–**F**) Upper panels: Representative western blots of the enzymes PDHK1, PDHK2, PDHK3, PDHK4, phosphorylated and total PDHE1*α* and LDHA. Lower panels: Densitometric quantification of the respective western blots after normalization to vinculin (**A**–**D**,**F**) or PDHE1*α* (E) (*n* = 4–6, * *p* < 0.05, LDHA: Mann–Whitney-Test; all other comparisons: unpaired Student’s *t*-test). All data represent mean + SD.

**Figure 5 cells-11-02213-f005:**
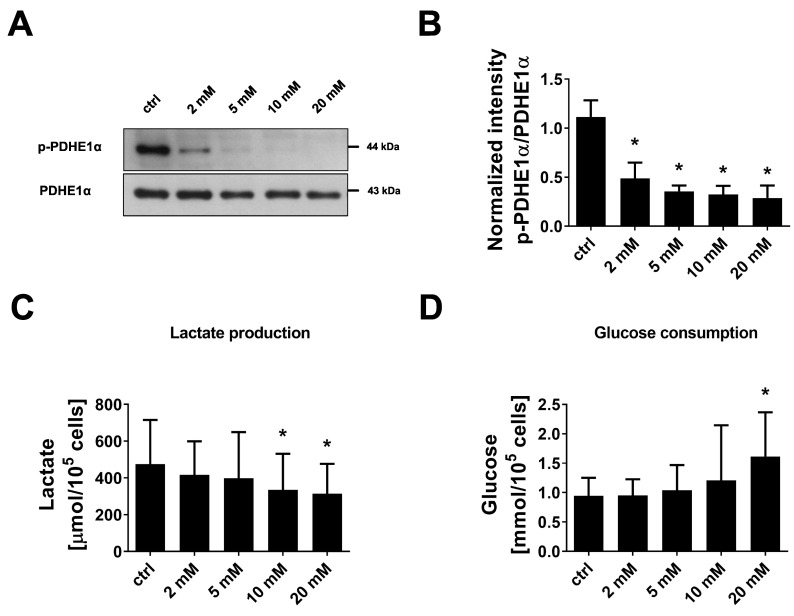
Dichloroacetate enhances glycolytic glucose metabolism in endothelial cells. Primary non-senescent HUVEC were left untreated (control, ctrl) or treated with 2–20 mM dichloroacetate (DCA) for 24 h. (**A**) Representative western blot of phosphorylated pyruvate dehydrogenase, subunit E1*α* (p-PDHE1*α*) at Ser293 and total PDHE1*α*. (**B**) Densitometric quantification of p-PDHE1*α* signals normalized to PDHE1*α* (*n* = 4, repeated measure (RM) one-way ANOVA, * *p* < 0.05 for post-hoc Holm-Šídák test). (**C**) Lactate levels in medium after 24 h incubation of cells in the presence or absence of DCA (*n* = 7, RM one-way ANOVA, * *p* < 0.05 for post-hoc Holm-Šídák test). (**D**) Glucose consumption as calculated from glucose levels in medium incubated for 24 h in the presence or absence of cells (*n* = 7, Friedman test, * *p* < 0.05 for post-hoc Dunn’s test). All data represent mean + SD.

**Figure 6 cells-11-02213-f006:**
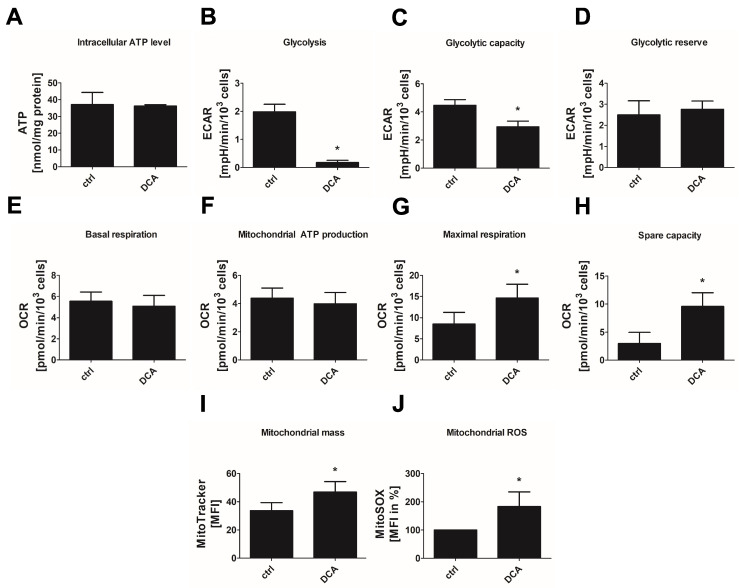
Dichloroacetate enhances glycolytic and oxidative glucose metabolism in endothelial cells. Primary non-senescent HUVEC were left untreated (control, ctrl) or treated with 20 mM of dichloroacetate (DCA) for 24 h. (**A**) Intracellular ATP level (*n* = 3, * *p* < 0.05, paired Student’s *t*-test). (**B**–**D**) Glycolysis stress test: glycolytic activity (**B**), capacity (**C**) and reserve (**D**) as calculated from extracellular acidification rates (ECAR) measured via Seahorse technology after sequential addition of glucose, oligomycin (inhibitor of ATP synthase) and 2-deoxyglucose (inhibitor of glycolysis) (*n* = 3, * *p* < 0.05, in (**B**,**C**): paired Student’s *t*-test, in (**D**): Wilcoxon matched-pairs signed-rank test). (**E**–**H**) Mito stress test: basal respiration (**E**), mitochondrial ATP production (**F**), maximal respiration (**G**) and spare capacity (**H**) as calculated from oxygen consumption rates (OCR) measured via Seahorse technology at baseline and after sequential addition of pharmacological agents (oligomycin, inhibitor of ATP synthase; carbonyl cyanide-4-(trifluoromethoxy)phenylhydrazone (FCCP), mitochondrial uncoupling agent; antimycin A, inhibitor of the respiratory chain complex III) (*n* = 4, * *p* < 0.05, paired Student’s *t*-test). (**I**) Mitochondrial mass depicted as relative median fluorescence intensity (MFI) of MitoTracker staining analyzed by flow cytometry (*n* = 4, * *p* < 0.05, paired Student’s *t*-test). (**J**) Mitochondrial reactive oxygen species displayed as relative MFI of MitoSOX analyzed by flow cytometry and normalized to control (*n* = 4, * *p* < 0.05, one-sample *t*-test). All data represent mean + SD.

**Figure 7 cells-11-02213-f007:**
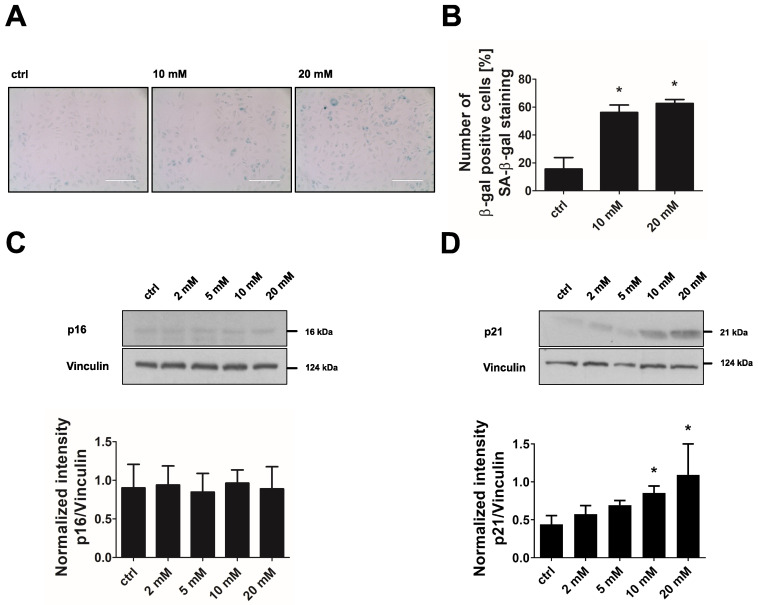
Dichloroacetate induces premature senescence. Primary non-senescent HUVEC were left untreated (control, ctrl) or treated with 2–20 mM of dichloroacetate (DCA) for 24 h. (**A**) Senescence-associated-β-galactosidase (SA-β-gal) staining. Scale bar = 200 µm. (**B**) Quantification of SA-β-gal-positive cells (*n* = 3, repeated measure (RM) one-way ANOVA, * *p* < 0.05 for post-hoc Holm-Šídák test). (**C**,**D**) Upper panels: Representative western blot of the senescence markers p16 and p21. Lower panels: Densitometric quantification of p16 and p21 after normalization to vinculin (*n* = 4, Friedman test: *p* < 0.05 for overall central tendency, * *p* < 0.05 for post-hoc Dunn’s test). All data represent mean + SD.

**Figure 8 cells-11-02213-f008:**
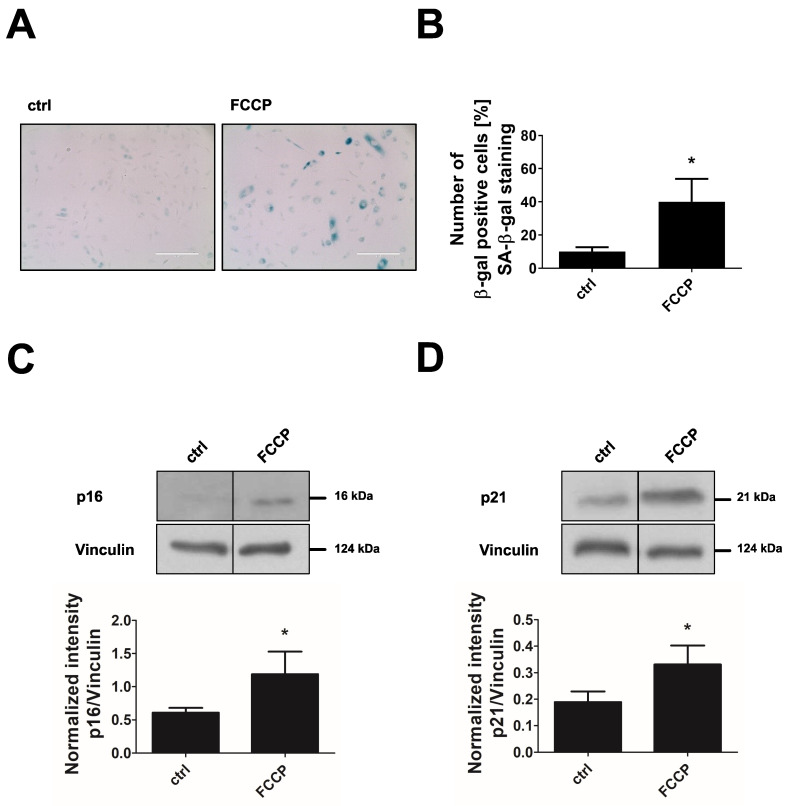
FCCP induces premature senescence. Primary non-senescent HUVEC were left untreated (control, ctrl) or treated with 5 µM of FCCP for 6 days. (**A**) Senescence-associated-β-galactosidase (SA-β-gal) staining. Scale bar = 200 µm. (**B**) Quantification of SA-β-gal-positive cells (*n* = 4, * *p* < 0.05, paired Student’s *t*-test). (**C**,**D**) Representative western blots of senescence markers p16 and p21 (upper panels) and densitometric quantification of p16 and p21 after normalization to vinculin (lower panels (*n* = 4, * *p* < 0.05, paired Student’s *t*-test). All data represent mean + SD.

## Data Availability

Not applicable.

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
