# Peer review of "Oxidative Glucose Metabolism Promotes Senescence in Vascular Endothelial Cells"

_cells, 2022, doi:10.3390/cells11142213_

Round 1

Reviewer 1 Report

This focused on glucose mechanism in endothelial aging. author provide sufficient data to support hypothesis and result is very interesting.

-Authors have analyzed passage 1 and 25 to compare cellular features. Indeed, replicative senescent cells showed the distinct different phenomenon in aspect of glucose usage or ROS productions. This difference is inferred to affect cell proliferation rate. In discussion, author mentioned that shift to oxidative metabolism by DCA was associated with inhibition of endothelial cell proliferation and migration. How about cell proliferation rate as PDT in p1 and p20 HUVEC? HUVEC is mostly available in vascular research and many research use HUVEC over passage 20. If author found the difference in cell division or shape in P1 and p25, researches should consider this alteration.

-In this study, glucose metabolism was chiefly dealt. How does author think about cytokine or growth factor production in replicative senescent cells? Oxidative stress is deeply associated with inflammation. 

-Please check panel in Figure  2. It's  missed.

Author Response

Point-to-point reply to the reviewer’s comments

We would like to thank the reviewers for the positive evaluation of our manuscript and the additional comments Below we address the reviewers’ comments in a point-to-point reply. New or altered text passages in the manuscript are highlighted in red.

Reviewer’s comments

Reviewer #1

Authors have analyzed passage 1 and 25 to compare cellular features. Indeed, replicative senescent cells showed the distinct different phenomenon in aspect of glucose usage or ROS productions. This difference is inferred to affect cell proliferation rate. In discussion, author mentioned that shift to oxidative metabolism by DCA was associated with inhibition of endothelial cell proliferation and migration.

We thank the reviewer for this positive judgment of our work.

Comment 1: How about cell proliferation rate as PDT in p1 and p20 HUVEC? HUVEC is mostly available in vascular research and many research use HUVEC over passage 20. If author found the difference in cell division or shape in P1 and p25, researches should consider this alteration.

We agree with the reviewer’s opinion that experiments with HUVEC are most meaningful if early passages are used. Passage 20-25 corresponds to 15-18 cumulative population doublings. We have added this information to the Materials and Methods section (line 150 – 151) and to the Results part (3.1., line 338 – 339).

Comment 2: In this study, glucose metabolism was chiefly dealt. How does author think about cytokine or growth factor production in replicative senescent cells? Oxidative stress is deeply associated with inflammation.

We thank the reviewer for this comment. Indeed, senescent cells are often found to exhibit a pro-inflammatory phenotype. We are currently studying cytokine and growth factor profiles in our model and, so far, found a distinct pattern in replicative senescent HUVEC compared to young cells with several inflammatory molecules being strongly upregulated while others are not changed or reduced. We plan to complete this work within a different project.

Comment 3: Please check panel in Figure 2. It's  missed.

We apologize for not having included panel labeling in Figure 2. We have now added the respective labeling (panel A – D).

Reviewer 2 Report

In this manuscript Stabenow et al. examined the role of PDH complex activity in promotion cellular senescence in vascular endothelial cells. They found that senescence cells showed higher glycolytic and TCA fluxes compared to young endothelial cells. They also found that increased PDHK expression could explain these phenotypes. They also found that pharmacological inhibition of PDHK in young endothelial cells promotes cell senescence. From these data, the authors concluded that alterations in glucose metabolism may act as a driving force for senescence in endothelial cells. Overall, I enjoyed reviewing this manuscript and found significant new findings in it. However, I do have some concerns regarding this paper.   Major:

1)    Title: The title should be reconsidered. This paper is well written and shows new findings in multiple methods. However, the title is not entirely consistent with the content of the Result section. Thus, the title should be rephrased to avoid misleading the reader.

2) Senescent endothelial cells showed both high glycolysis flux and PDH flux. As the Authors stated in the Introduction, inhibition of PDHK stimulates PDH flux but decreases glycolytic flux. The Authors should explain why this happens. LDH activity alone is not sufficient to produce lactate. Cytoplasmic NAD concentration may explain this phenomenon.

3) I think experiments with DCA needs to be improved. As reported, DCA has an inhibitory effect on PDHK. However, it is assumed that there are additional effects on the HUVEC. At least, Authors should add dose-response experiments with phospho-PDH and some of the outcomes such as ECAR, glucose consumption, and lactate production. In addition, another PDHK inhibitor should be applied to these experiments.

4) SiRNA against PDHK may strengthen their conclusions.

5) Critical point. I love Author’s hypothesis, “metabolic change enhance cell senescence”. To prove this hypothesis other methods of switching PDH flux need to be applied. For example, uncoupler such as FCCP stimulate PDH flux; does FCCP enhance cellular senescence? In contrast, 2-DOG decrease both glycolysis and PDH flux; does 2-DOG inhibit the senescence-promoting effects of DCA?

6) In Figure 7: Western blot of p16 should be added.

Author Response

Point-to-point reply to the reviewer’s comments

We would like to thank the reviewers for the positive evaluation of our manuscript and the additional comments Below we address the reviewers’ comments in a point-to-point reply. New or altered text passages in the manuscript are highlighted in red.

Reviewer’s comments

Reviewer #2

In this manuscript Stabenow et al. examined the role of PDH complex activity in promotion cellular senescence in vascular endothelial cells. They found that senescence cells showed higher glycolytic and TCA fluxes compared to young endothelial cells. They also found that increased PDHK expression could explain these phenotypes. They also found that pharmacological inhibition of PDHK in young endothelial cells promotes cell senescence. From these data, the authors concluded that alterations in glucose metabolism may act as a driving force for senescence in endothelial cells. Overall, I enjoyed reviewing this manuscript and found significant new findings in it. However, I do have some concerns regarding this paper.

We thank the reviewer for this positive comment and appreciate the constructive suggestions.

Comment 1: Title: The title should be reconsidered. This paper is well written and shows new findings in multiple methods. However, the title is not entirely consistent with the content of the Result section. Thus, the title should be rephrased to avoid misleading the reader.

We have modified the title of the manuscript by referring to oxidative glucose metabolism. The new title is “Oxidative glucose metabolism promotes senescence in vascular endothelial cells”.

Comment 2: Senescent endothelial cells showed both high glycolysis flux and PDH flux. As the Authors stated in the Introduction, inhibition of PDHK stimulates PDH flux but decreases glycolytic flux. The authors should explain why this happens. LDH activity alone is not sufficient to produce lactate. Cytoplasmic NAD concentration may explain this phenomenon.

We thank the reviewer for this comment. Indeed, we show that both, replicative senescent endothelial cells characterized by reduced expression of all PDHK isoforms, and cells, in which PDHKs were pharmacologically inhibited by DCA, exhibited an increase of oxidative glucose metabolism. From these results and from the experiments where premature senescence was induced by increasing PDH flux via uncoupling the respiratory chain (see below), we concluded that increased oxidative metabolism promotes senescence. However, while glycolysis was enhanced in replicative senescence, it was decreased in DCA-induced premature senescence as previously reported (Harting et al. 20171). We think that this difference is due to the more complex metabolic changes in replicative senescent cells where not only PDHK isoforms are downregulated but also LDHA is upregulated. LDHA generates lactate and recycles the NADH generated in glycolysis back to NAD+, which is required for the action of glyceraldehyde-3-phosphate dehydrogenase, a key enzyme in glycolysis. Thus, glycolysis can only be maintained or be increased if the NAD+ supply is adequate. An enhanced PDH activity without concomitant LDHA upregulation may not only promote oxidative pyruvate metabolism but additionally limit NAD+ availability and thus, glycolysis. In fact, LDHA protein expression was not changed in DCA-treated cells in our study (see supplementary figure 2 included into the revised manuscript). Of note, a previous study has shown that PDHK inhibition led to a reduced NAD+/NADH ratio (Luengo et al 20212). We have now added this point to the Discussion (lines 758 – 761, 772 – 776).

 1 Harting, T.P.; Stubbendorff, M.; Hammer, S.C.; Schadzek, P.; Ngezahayo, A.; Murua Escobar, H.; Nolte, I. Dichloroacetate affects proliferation but not apoptosis in canine mammary cell lines. PLoS One 2017, 12, e0178744, doi:10.1371/journal.pone.0178744.

2 Luengo A, Li Z, Gui DY, Sullivan LB, Zagorulya M, Do BT, Ferreira R, Naamati A, Ali A, Lewis CA, Thomas CJ, Spranger S, Matheson NJ, Vander Heiden MG. Increased demand for NAD+ relative to ATP drives aerobic glycolysis. Mol Cell. 2021 Feb 18;81(4):691-707.e6. doi: 10.1016/j.molcel.2020.12.012.

Comment 3: I think experiments with DCA needs to be improved. As reported, DCA has an inhibitory effect on PDHK. However, it is assumed that there are additional effects on the HUVEC. At least, Authors should add dose-response experiments with phospho-PDH and some of the outcomes such as ECAR, glucose consumption, and lactate production. In addition, another PDHK inhibitor should be applied to these experiments.

We thank the reviewer for this suggestion. We have now added dose-response experiments for DCA treatment (2 – 20 mM, 24 h) and show that inhibition of pyruvate dehydrogenase, subunit E1α (PDHE1α) phosphorylation at Ser293, an indicator for PDHK inhibition, as well as reduction of lactate production and increase of glucose consumption occurred in a dose-dependent manner. Moreover, the DCA-induced premature senescence of endothelial cells as shown by increased expression of p21 was dose-dependent although significance was only reached with DCA concentration of 10 mM and 20 mM. Similarly, an increase of SA-β-gal-positive cells was seen after treatment of 10 mM or 20 mM. We did not detect an increase of p16, another senescence marker, which corresponds to observation in other premature senescence models (for instance, senescence induced by oxidative stress, our own data). We have included these new data into the Method part (Lines 157, 200, 218, 326 – 331), the Results section (line 480 – 485, 568 - 571) and in Figure 5A-D and Figure 7A-D as well as into the Discussion (Lines 735, 753 – 754).

As suggested by the reviewer, we have also performed experiments with another PDHK inhibitor, KPLH1130, which was recently described to be functional in macrophages (Min et al., 20191). Unfortunately, applying this inhibitor to endothelial cells according to the previously published conditions did not lead to inhibition of PDHE1α phosphorylation and increasing dose and incubation time led to reduction of cell viability. However, we followed the suggestion of the reviewer to use a second approach to increase PDH flux by uncoupling mitochondrial respiratory chain and showed that this also led to premature senescence (see comment 5).

For figure, please, check the PDF file of the rebuttal letter

Effect of KPLH11301 on PDHE1α phosphory-lation in endothelial cells. Human umbilical vein endothelial cells were treated with different concentrations of KPLH1130 for 24 h, lysed and subjected to immunoblot analysis of phosphorylated (Ser293) and total pyruvate dehydrogenase, subunit E1α (p-PDHE1α).

1 Min BK, Park S, Kang HJ, Kim DW, Ham HJ, Ha CM, Choi BJ, Lee JY, Oh CJ, Yoo EK, Kim HE, Kim BG, Jeon JH, Hyeon DY, Hwang D, Kim YH, Lee CH, Lee T, Kim JW, Choi YK, Park KG, Chawla A, Lee J, Harris RA, Lee IK. Pyruvate Dehydrogenase Kinase Is a Metabolic Checkpoint for Polarization of Macrophages to the M1 Phenotype. Front Immunol. 2019 May 7;10:944. doi: 10.3389/fimmu.2019.00944. PMID: 31134063; PMCID: PMC6514528.

Comment 4: SiRNA against PDHK may strengthen their conclusions.

We agree with the reviewer that siRNA-mediated knockdown of PDHK would have strengthened our conclusion. Indeed, we have applied siRNAs against PDHK1 and 4, the isoforms, which were mainly described in endothelial cells. So far, we were able to successfully downregulate PDHK1 but not PDHK4. However, downregulation of PDHK1 was not accompanied by inhibition of pyruvate dehydrogenase, subunit E1α (PDHE1α) phosphorylation suggesting that other PDHK isoforms compensate the effect of PDHK1 downregulation. Thus, we would probably need to downregulate all 4 PDHK isoforms by applying 4 different siRNAs, which is technically challenging, would lead to accumulation of off-target effects and make interpretations of the results difficult.

For figure, please, check the PDF file of the rebuttal letter

Effect of siRNA-mediated downregulation of PDHK1 on PDHE1α phosphorylation in endothelial cells. Human umbilical vein endothelial cells were treated with specific siRNA against PDHK1 (1 µg, 96 h or 96 h with re-transfection for 96 h (2x96 h), lysed and subjected to immunoblot analysis of PDHK1, vinculin (loading control), phosphorylated (Ser293) and total pyruvate dehydrogenase, subunit E1α (p-PDHE1α). Two biological replicates per condition are shown.

Comment 5: Critical point. I love Author’s hypothesis, “metabolic change enhance cell senescence”. To prove this hypothesis other methods of switching PDH flux need to be applied. For example, uncoupler such as FCCP stimulate PDH flux; does FCCP enhance cellular senescence? In contrast, 2-DOG decrease both glycolysis and PDH flux; does 2-DOG inhibit the senescence-promoting effects of DCA?

We thank the reviewer for this valuable suggestion. We have now performed experiments with FCCP (5 µM, 6 days) and show that this treatment is a strong inducer of premature senescence in endothelial cells. This was proven by increased of SA-β-gal-positive cells and an increase of the two senescence markers p21 and p16. These data present additional evidence for linking metabolic changes to senescence. We have included these new data into the Method section (line 157 – 158), the Results section (line 573 – 579) and in Figure 8A-D, and refer to them in the Discussion section (line 762 – 764).

As far as the second question of the reviewer is concerned, the prevention of DCA-induced senescence by 2-DOG, we want to mention that a long-term treatment with DCA (24 h) was needed to see premature senescence. Accordingly, we would need to have an at least equally long incubation with 2-DOG, which would lead to energy depletion, in part counteracted by increased fatty acid oxidation, and apoptosis, which would complicate the interpretation of these results. Thus, we have not obtained experimental data here.

Comment 6: In Figure 7: Western blot of p16 should be added.

We apologize not to have shown p16 western blots for DCA-treated samples in the previous version of this manuscript. We have included these data now (Result section line 570 – 571, Figure 7C). DCA-induced premature senescence is not associated with increased p16 expression, which corresponds to other models of stress-induced senescence in endothelial cells (for instance oxidative stress-induced senescence, our own data). However, dependent on the extent and duration of cellular stress, p16 can also be upregulated as shown in the FCCP model of premature senescence.

Round 2

Reviewer 2 Report

I have no further comment.